# RUNNING HUGE CONTEXT WINDOWS ON TINY GPUS

## ABSTRACT

There is growing demand for large language models which can process hundreds of thousands or even millions of input tokens. Inference at this extreme scale demands significant computational resources and costs. To address the inference time costs associated with running self-attention based transformer language models on long contexts, we propose a tunable mechanism that reduces the cost of the forward pass by attending to only the most relevant tokens at every generation step using a top-k selection mechanism. We showcase the efficiency gains afforded by our method by performing inference on context windows up to 1M tokens using approximately 16GB of GPU RAM. Our experiments reveal that models are capable of handling the sparsity induced by the reduced number of keys and values. By attending to less than 1% of input tokens, we achieve over 95% of model performance on common long context benchmarks (LM-Eval, AlpacaEval, and RULER).

## 1 INTRODUCTION

Long-context inference enables models to analyze large document collections or follow long and detailed instructions. During language model inference, tokens are produced one token at a time, with each new token attending to every token in the context window. As a result, long-context inference is expensive. To solve this, brute-force approaches to long-context in Large Language

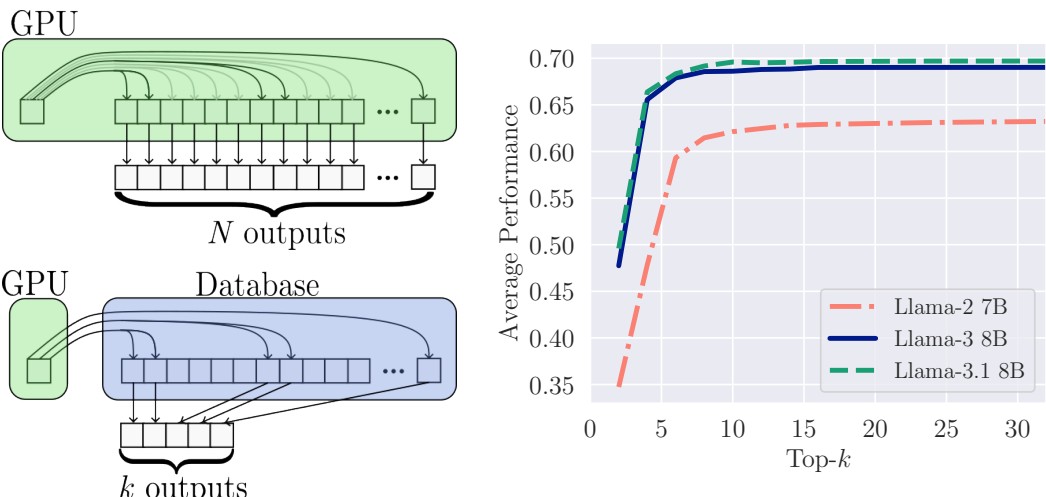

Figure 1: (left-top) Typical attention requires each query vector to attend over—compute an inner product with—each key vector in the context window. In practice, very few key vectors contribute significantly to attention, and much of this computation is wasted. (left-bottom) top-$k$ attention retrieves only the keys that contribute significantly to the attention computation, leaving the gray arrows out and achieving sublinear runtime. (right) Performance on OpenLLM Leaderboard using only the top-$k$ keys for each attention computation. Typical questions have a context length of $\sim 1000$, yet only 10 keys are needed to achieve the same performance as full attention.

Models (LLMs) are commonplace. Leading this charge is Ring Attention (Liu et al., 2023), which uses round-robin communication between many servers to scale up computation. However, this approach is not a sustainable or commercially viable long-term solution due to the extremely high per-token inference costs.

In this paper, we observe that modern LLMs only require the top contributors to each attention computation to perform well. This opens the door to do long-context inference with very little GPU computing power. We build an implementation of attention in which keys and values are stored in a vector database in CPU memory. When attention is computed using a query vector, we retrieve only the keys with the largest attention scores. This retrieval can be done in sublinear time using an approximate k-nearest neighbor search, enabling long-context inference using plentiful and cheap CPU memory, and without the heavy computational overhead required for full attention.

Our proposed approach bears similarities to Retrieval Augmented Generation (RAG), in which a large corpus of text is analyzed by retrieving a small subset of the most relevant documents from a vector database and placing them into the context window. However, retrieval models lack the sophisticated zero-shot and reasoning capabilities of a full-scale language model. Our proposed approach is a middle ground between long-context inference and RAG; we retain the inherent capabilities of language models, but also leverage a vector database to reduce computation. By seamlessly integrating the capabilities of long-context inference and RAG, we make it possible to perform generation over context lengths in the millions of tokens with a single, inexpensive GPU.

Our primary contributions are summarized as follows:

- We propose a simple method for sublinear long-context generation by using top-$k$ attention over preprocessed states in a vector database.

- We show that this technique achieves high fidelity on common benchmarks and long-context tasks.

- We provide empirical analysis on why top-$k$ attention works and prescriptive recommendations for choosing the optimal $k$ for a given task.

## 2 MOTIVATION

Our work on long context inference is motivated by the observation that modern language models naturally have extremely sparse attention patterns in which a very small number of tokens make up a the vast majority of attention mass.

We visualize the sparsity of attention in Llama-3-8B using 50 Wikipedia articles in Figure 2. For the last token in each 4000-token context window, we tabulate the number of keys in the context that is needed to collect 75% of the attention mass. We see that a relatively small number of the 4000 tokens are needed to collect this mass, especially for deeper layers of the networks.

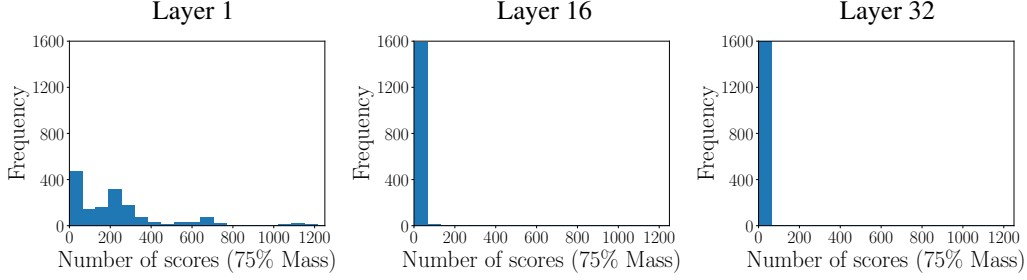

Figure 2: We analyze the number of attention scores that correspond to 75% of the probability mass for generating the next token. Each point is the number of scores of the last 'row' from the attention matrix required to reach 75% of the total attention. We observe each of the 32 heads across 50 samples. Each sample consists of 4000 tokens but no sample requires more than 1250 tokens to account for 75% of the attention.

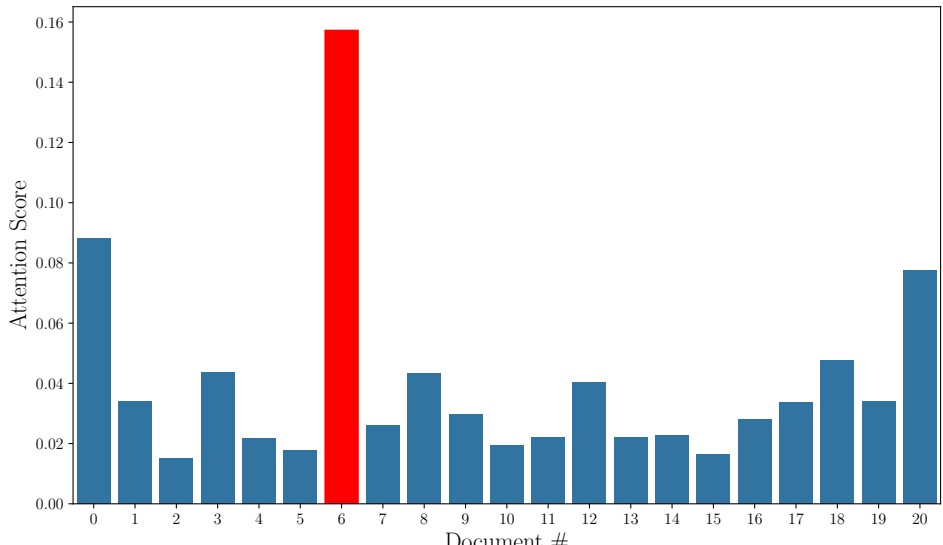

Figure 3: Where does the attention go across multiple documents? Observing all of the attention scores over all of the heads and all of the layers we see that in expectation the model is able focus most of its attention on the correct document.

Next, we take our multi-document long context samples and ask Llama to copy one of the documents that relates to a specific topic. Figure 3 demonstrates that in expectation the attention scores (across all heads and layers) are higher for the document containing the correct document.

To further analyze the sparsity of the model, we observe the entropy of the attention weights for each token as a function of layer depth. Figure 4 shows that entropy is low across all layers, and drops significantly after the first layer.

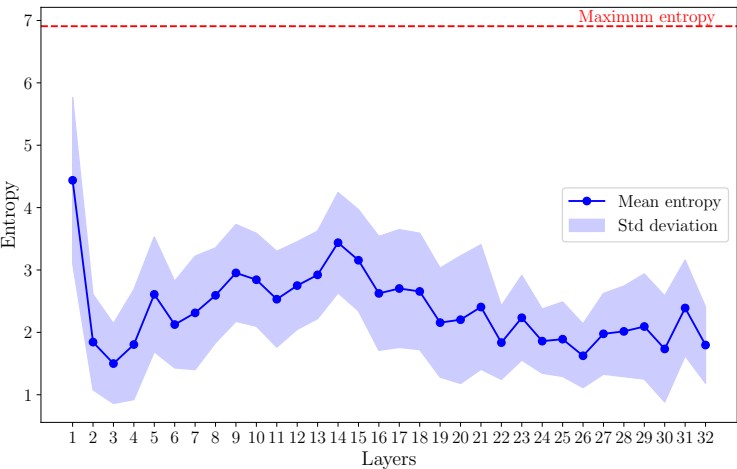

Figure 4: Entropy of the distribution of attention scores for each layer of a model. Attention score distributions are derived from 50 samples and aggregated over all heads for a given layer. Entropy serves as a measure of how concentrated the attention scores are for a given query token: low entropy indicates a large amount of attention centered over few tokens, and high entropy indicates a more uniform dispersion of attention scores.

The sparsity in Llama's attention weights suggests that few key-value pairs should be needed for the model to perform close it its full capabilities. We test this hypothesis in Figure 1 (right) by evaluating models from the Llama family across varying quantities of key-value pairs. In these experiments, we

use the same number of keys for each layer of the network and average the model performance over all tasks on the OpenLLM leaderboard. We see that all three models saturate in performance by the 15th key, and the most recent (and most overtrained) variants of the model require only 10 top keys to achieve the same performance as full-scale attentions. While Figure 2 indicates that there is some spreading of the attention mass for layer 1 of the network, this tail mass seems to be unnecessary for good performance on benchmarks.

These experiments motivate the idea of exploiting model sparsity to speed up attention computations. Empirically, very few keys are needed to perform inference, but it is important to select keys that are primary contributors to the attention computation. The clear way forward is to use a vector database to retrieve the top-k most influential tokens, enabling the GPU to perform matrix computations while the keys and values live in CPU memory. This approach alleviates both the computational and memory barriers to doing long-context inference on a small GPU.

## 3 METHODOLOGY

In this section, we elaborate on our method for reducing memory costs at inference time using top-$k$ attention. But first, we introduce some notation and common terms related to attention blocks.

### 3.1 CAUSAL SELF-ATTENTION

The first step of self-attention as implemented in a standard transformer block is to project the sequence of token embeddings $x \in \mathbb{R}^{N \times D}$ via learned weight matrices into queries, keys, and values: $Q = xW_Q, K = xW_K, V = xW_V \in \mathbb{R}^{N \times D}$. The canonical scaled softmax attention is implemented by "scoring" the relevance of each key vector to each query vector via inner products, and then the resulting "attention" scores are renormalized via a softmax function. Finally, at each query position, the scores are used to weight a summation over value vectors to produce the output hidden state at every position. This is often compactly represented using the following matrix notation:

$$\text{Attention}(Q, K, V) = \text{Softmax}\left(\frac{QK^T}{\sqrt{D}}\right) V. \tag{1}$$

We refer to $S = QK^T$ as the *score* matrix, and $A = \text{softmax}(S/\sqrt{D})$ as the *attention* matrix. The transformer block performs a few more operations including normalization and position-wise feedforward transformations, although our implementation leaves these operations intact and we do not consider them further.

### 3.2 INFERENCE WITH KV CACHES

Autoregressive transformer language models generate text by repeatedly predicting the most likely next token conditioned on a sequence of previous tokens, and then concatenating the newly chosen token to the original input sequence before repeating the process. Because the key and value computations of past tokens are unaffected by the addition of future tokens, it is typical to cache key and value pairs from each token and re-use them each time a token is generated.

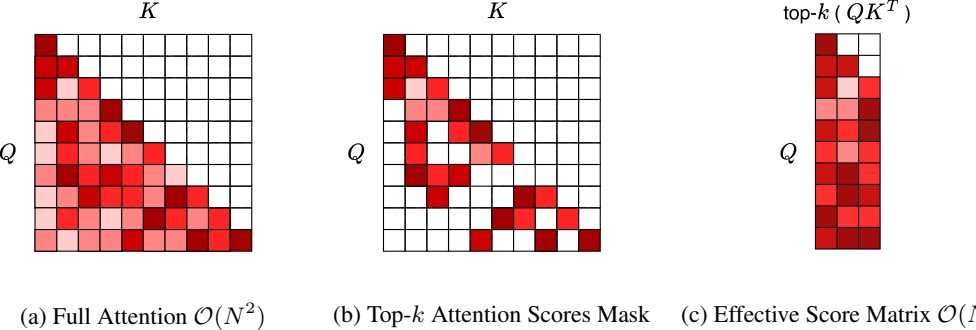

(a) Full Attention $\mathcal{O}(N^2)$    (b) Top-$k$ Attention Scores Mask    (c) Effective Score Matrix $\mathcal{O}(Nk)$

Figure 5: Comparing dense causal self-attention vs top-$k$ causal attention with $k = 3$.

When keys and values are cached for all tokens in the context window, a token can be generated by producing the query, key, and value vectors for the latest token, and compare this new query vector against all previous keys.

Since the single new query is a vector $q_i \in \mathbb{R}^{1 \times D}$, when multiplied by all previous $N + N_{gen}$ keys in the cache during attention, the memory cost incurred for the score matrix at this step is now only $O(N)$. This process is formalized in Algorithm 1.

---

**Algorithm 1** Standard KV Cache Generation

---

**Require:** K_cache $= \{K_\ell\}_{\ell=1}^L$, V_cache $= \{V_\ell\}_{\ell=1}^L$, token $x \in \mathbb{R}^{1 \times D}$, $N_{max} \in \mathbb{N}$
1: $N_{gen} = 1$
2: **while** $N_{gen} < N_{max}$ **do**
3:     **for** $\ell \in \{1, \ldots, L\}$ **do**
4:         $Q = xW_Q, K = xW_K, V = xW_V$           $\triangleright Q, K, V \in \mathbb{R}^{1 \times D}$
5:         K_cache$[\ell] \leftarrow$ concat(K_cache$[\ell], K$)     $\triangleright$ K_cache$[\ell] \in \mathbb{R}^{(N+N_{gen}) \times D}$
6:         V_cache$[\ell] \leftarrow$ concat(V_cache$[\ell], V$)     $\triangleright$ V_cache$[\ell] \in \mathbb{R}^{(N+N_{gen}) \times D}$
7:         $S \leftarrow Q \cdot$ K_cache$[\ell]^T$             $\triangleright S \in \mathbb{R}^{1 \times (N+N_{gen})}$
8:         $\hat{x} \leftarrow$ softmax$\left(\frac{S}{\sqrt{D}}\right) \cdot$ V_cache$[\ell]$         $\triangleright \hat{x} \in \mathbb{R}^{1 \times D}$
9:     **end for**
10:    $x \leftarrow$ sample_new_token($\hat{x}$)
11:    $N_{gen} \leftarrow N_{gen} + 1$
12: **end while**

---

### 3.3 TOP-$k$ ATTENTION

Even when using a KV Cache, as the length of the sequence grows, the cost of computing the attention scores for just the single newly generated token becomes non-trivial. We cut down on this growing cost by only considering a subset of the most relevant keys in the KV cache by performing a k-nearest neighbor search over the key vectors for the new query vector, returning only those value vectors corresponding to the $k$-largest attention scores:

$$\text{top-}k\text{-Attention}(Q, K, V) = \text{Softmax}\left(\frac{1}{\sqrt{D}}\text{top-}k\left(QK^T\right)\right)V. \tag{2}$$

This can be visualized as a mask selecting only the $k$ largest values in each row of the attention matrix (Figure 5 center). This increases the sparsity of the attention matrices while retaining the most important score values. The weighted sums of value vectors are taken only over the $k$ value vectors corresponding to the $k$ largest attention scores, a final operation with complexity only $\mathcal{O}(k)$.

An optimal implementation of this method abstracts the implementation and, critically, space-time complexity of the $k$-NN search over the KV cache. In our experiments we offload the KV cache and vector search operation to the CPU while performing the $Q$, $K$, and $V$ projections on the GPU (as well as other matrix operations in the transformer model).

Algorithm 2 concretely describes how our method takes in a KV Cache and constructs a nearest neighbor vector search data structure with the keys as the "database" vectors. When a new query vector is generated, it searches over the cache for the top-$k$ score values, and densely computes the score values with respect to any previously generated tokens. $K$ and $V$ projections for the newly generated tokens can optionally be added to the database. However, our experiments operate on the scale of millions for the database and thousands for generated tokens, so we maintain a dense KV cache for newly generated tokens on the GPU, while using a sparse top-k cache for keys in the pre-computed context cache.

### 3.4 CONSTRUCTING A KV CACHE AT THE MILLION TOKEN SCALE

The initial forward pass in the construction of a KV cache potentially requires incurring the full $O(N^2)$ memory cost. There are many ways this could be done. Given access to large amounts of

---

**Algorithm 2** Top-$k$ KV Cache Generation

---

**Require:** `K_cache` $= \{K_\ell\}_{\ell=1}^L$, `V_cache` $= \{V_\ell\}_{\ell=1}^L$, token $x \in \mathbb{R}^{1\times D}$, $k \in \mathbb{N}$, $N_{max} \in \mathbb{N}$

1: $N_{gen} = 1$, `K_cache_gen` $= []$, `V_cache_gen` $= []$
2: `K_cache` $\leftarrow$ `build_knn(K_cache)`
3: **while** $N_{gen} < N_{max}$ **do**
4:   **for** $\ell \in \{1, \ldots, L\}$ **do**
5:     $Q = xW_Q, K = xW_K, V = xW_V$                            $\triangleright Q, K, V \in \mathbb{R}^D$
6:     `K_cache_gen`$[\ell] \leftarrow$ `concat(K_cache_gen`$[\ell], K)$  $\triangleright$ `K_cache_gen`$[\ell] \in \mathbb{R}^{N_{gen}\times D}$
7:     `V_cache_gen`$[\ell] \leftarrow$ `concat(V_cache_gen`$[\ell], V)$  $\triangleright$ `V_cache_gen`$[\ell] \in \mathbb{R}^{N_{gen}\times D}$
8:     `vals, idx` $\leftarrow$ `topk_query`$(Q,$ `K_cache`$, k)$          $\triangleright V, I \in \mathbb{R}^{1\times k}$
9:     $A \leftarrow$ `Softmax(construct_sparse_matrix`$(V, I))$    $\triangleright A \in \mathbb{R}^{1\times N}, A = O(k)$
10:    $A_{gen} \leftarrow$ `Softmax`$\left(\frac{1}{\sqrt{D}}Q \cdot$ `K_cache_gen`$[\ell]^T\right)$        $\triangleright A_{gen} \in \mathbb{R}^{1\times N_{gen}}$
11:    $\hat{x} \leftarrow AV + A_{gen}$`V_cache_gen`$[\ell]$
12:   **end for**
13:   $x \leftarrow$ `sample_new_token`$(\hat{x})$
14:   $N_{gen} \leftarrow N_{gen} + 1$
15: **end while**

---

compute, one could parallelize over many GPUs and construct the cache using algorithms like Ring Attention Liu et al. (2023). If such hardware is not available, approximate algorithms like windowed attention would allow for the construction of large caches with more modest compute Child et al. (2019). For small enough $N$ (100's of thousands of tokens), and with a high-memory GPU, the vLLM library can quickly construct KV caches by performing a standard forward pass on the model using the paged attention algorithm Kwon et al. (2023). Finally, for very large N, top-$k$ attention could be employed at cache construction time as well. In our experiments, we employ a variety of these techniques depending on the model and context window size.

## 4    EVALUATING TOP-$k$ ATTENTION AT SCALE

We evaluate top-$k$ on various benchmarks to highlight the relationship between different $k$ values and performance on various benchmarks. We find that models of various sizes and generations perform well even at small $k$. In general, we observe that a $k$ equal to 1% of the total context length is sufficient to achieve 95% of the performance achieved with full, standard attention.

### 4.1    EVALUATIONS

We analyze the top-$k$ attention mechanism's effectiveness across three benchmarks: AlpacaEval, Open LLM Leaderboard v1 tasks, and RULER. Each of these benchmark highlight a different quality. RULER tests the model's long context capability. AlpacaEval measures generation quality of the model. Open LLM Leaderboard v1 tasks test the model's capabilities such as knowledge. Measuring performance across these datasets presents a comprehensive understanding of top-$k$.

**RULER**   To demonstrate that top-$k$ attention with small $k$ remains effective as the context length increases, we run the RULER (Hsieh et al., 2024) benchmark suite over a series of increasing context lengths. As shown in Table 1 we run over lengths from 4k to 128k. RULER consists of thirteen total tasks from four categories. The evaluation harness runs the original Needle In A Haystack (NIAH) (Kamradt, 2023) task along with a series of more challenging variations on the task. (For example in one task the text consists entirely of labeled "needles" and the model is queries to retrieve the needle corresponding to a single label.) These NIAH tasks comprise 8 of the 13 tasks, and the remaining tasks are split into three categories: summarization proxies, multi-hop proxies, and question answering.

**AlpacaEval 2.0**   We benchmark top-$k$ attention on AlpacaEval (Dubois et al., 2024). AlpacaEval 2.0 requires a model to generate responses to 805 queries. These responses are then compared

Table 1: Results on RULER benchmark at various context lengths. Scores represent an average of 13 tasks in the RULER benchmark, with maximum possible score being 100. The RULER benchmark was run separately for each context length listed, and each context length was run with top-$k$ attention for increasing values of $k$ and also with standard, full attention.

| $k$ | Context Length (tokens) | | | | | |
|---|---|---|---|---|---|---|
| | 4096 | 8192 | 16384 | 32768 | 65536 | 131072 |
| **2** | 70.42 | 67.82 | 75.36 | 67.95 | 67.83 | 64.76 |
| **8** | 87.44 | 80.01 | 86.69 | 84.56 | 75.99 | 61.28 |
| **32** | 88.35 | 84.35 | 88.08 | 84.56 | 78.18 | 65.89 |
| **128** | 88.85 | 86.86 | 89.08 | 84.56 | 77.41 | 73.59 |
| **512** | 88.73 | 88.10 | 89.31 | 84.56 | 77.41 | 63.58 |
| **2048** | 90.04 | 88.77 | 89.31 | 84.56 | 77.33 | 64.62 |
| **8192** | – | – | 88.85 | 84.56 | 77.41 | 58.53 |
| **16384** | – | – | – | 83.79 | 77.26 | 73.40 |
| **32768** | – | – | – | – | 78.03 | 73.40 |
| **Full** | 92.35 | 89.27 | 89.12 | 84.56 | 78.87 | 75.17 |

by a LLM-as-a-Judge to GPT-4 Turbo responses generated from the same query set. The winrate percentage is reported, and the LLM-as-a-Judge is GPT-4 Turbo.

**Open LLM Leaderboard Tasks**  We investigate the performance of top-$k$ on Open LLM Leaderboard tasks. Particularly, we evaluate different models on various values of $k$ on the average of MMLU (Hendrycks et al., 2020), ARC tasks both easy and challenge (Clark et al., 2018), HellaSwag (Zellers et al., 2019), winogrande (Keisuke et al., 2019), OpenbookQA (Mihaylov et al., 2018), BoolQ (Clark et al., 2019), and PiQA (Bisk et al., 2020). For each task, we record the normalized accuracy when available; otherwise, we record accuracy. We report the average over tasks. We evaluate the following models on these benchmarks: llama-1 (7B), llama-2 (7B), llama-3 (8B), llama-3.1 (8B), Vicuna-v1.3 (7B), llama-2 chat (7B), llama-3 Instruct (8B), llama-3.1 instruct (8B), llama-3.2 1B instruct, and llama-3.2 3B instruct (Touvron et al., 2023a; Zheng et al., 2023; Touvron et al., 2023b; Dubey et al., 2024; AI, 2024). The experiments are conducted using `lm-eval-harness` in a zero-shot setting (Gao et al., 2024).

## 4.2  TOP-$k$ IS EFFECTIVE AT LOW $k$

**Top-$k$ Performance on RULER**  We evaluate RULER using GradientAI's Llama-3-8B model that has a trained context length of 262k. We find that very small values of $k$ are sufficient to recover near-baseline performance. For every context length evaluated, $95\%$ of the baseline performance can always be achieved with a $k$ value of $1\%$ or less of the total length. In Table 1, at $k = 2$, greater than $60\%$ performance is achieved at all context lengths. The performance on RULER improves as $k$ increases. Nevertheless, even at a context length of 131k tokens to achieve $\sim 98\%$ performance of the full attention only $12.5\%$ of the attention scores are required. This highlights the effectiveness of top-$k$ on long-context.

Interestingly, most mechanical tasks have high success rates with little variation across runs. Conversely, the question-answering tasks (from SQuAD (Rajpurkar et al., 2016) and HotpotQA Yang et al. (2018)) end up being the tasks most indicative of model capability, with a consistently decreasing score as context length increases and as $k$ decreases. These QA tasks were both the most revealing and had the highest variance across test runs, so allocating an extra compute budget to this subset of tasks would be helpful.

**Top-$k$ Performance on AlpacaEval**  Of the three benchmarks evaluated, AlpacaEval required the largest $k$ as a percentage of context length to achieve $95\%$ of baseline performance, with $2.5\%$ of the context length being required. The trend of small $k$ values achieving near-baseline performance was repeated across model sizes and generations.

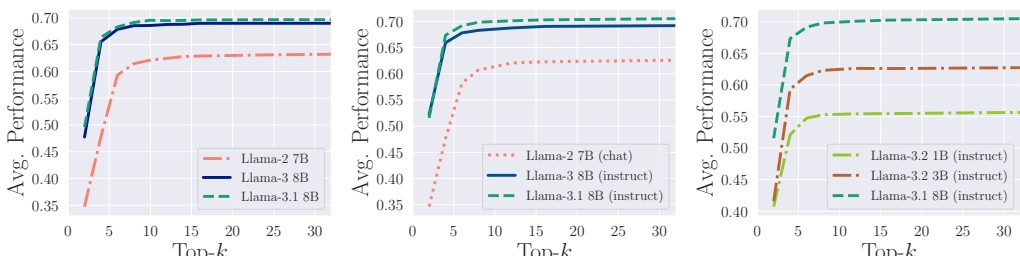

Figure 6: Top-K attention is effective for OpenLLM Leaderboard Tasks even at small values of $k$. Left shows the average of all tasks as we increase $k$ on pretrained base models. Center shows instruction tuned models. Right investigates the performance on different model sizes.

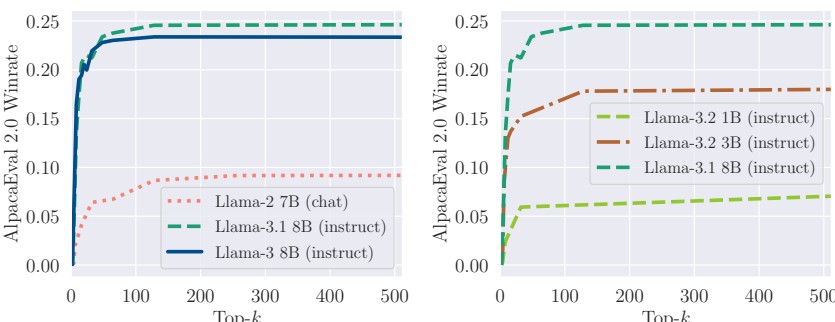

Figure 7: AlpacaEval 2.0 results for various models showing what value of $k$ is required to achieve the same benchmark performance as standard, full attention. Left compares different generations of llama instruction tuned models. Right investigates how models of different sizes handle small values of $k$.

**Top-$k$ Performance on OpenLLM Leaderboard Tasks**  We evaluated various models to find that top-$k$ behavior exists regardless of instruction tuning, model size, or the number of tokens the model was trained on. When comparing the models with different amounts of tokens trained, Table 6 left shows all models exhibit a similar curve with performance quickly saturating by a $k$ value $< 10$, regardless of the number of tokens on which the model was trained. Furthermore, when comparing Table 6 left and center, we see that instruction models and pretrained base models exhibit the same behavior, very quickly saturating. Additionally, regardless of model size, in Table 6 right, we see that whether a model is 1B, 3B, or 8B the behavior of top-$k$ is the same.

## 4.3 HUGE CONTEXT GENERATION WITH TOP-$k$

To demonstrate the scaling that top-$k$ attention permits, we use our method to generate tokens from a context window of one million tokens. We choose RULER's Needle In A Haystack task to use for the context, and we use GradientAI's Llama-3-8B model that has a trained context length of 1M (Gradient Team, 2024). We run this on a single-GPU node with a Faiss vector database containing the KV cache.[1] We also compare our method against Xiao et al. (2023) for needle in a haystack in figure 8.

---

[1] We used a variety of $k$ values for this experiment and it turned out that $k$=1 is sufficient to solve Needle In A Haystack with a context length of 1 million tokens.

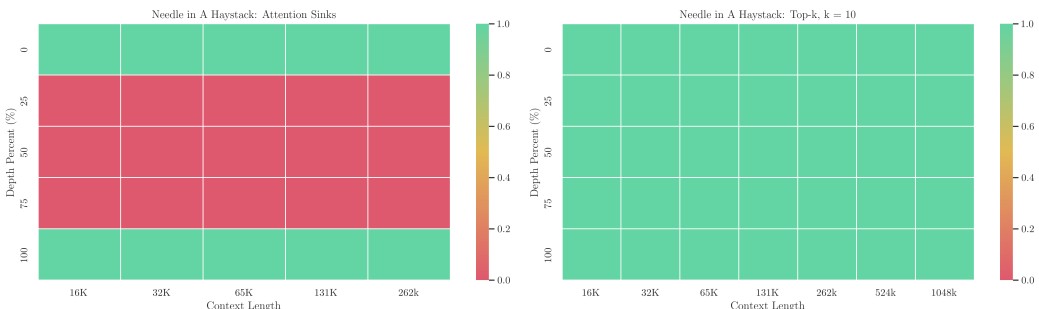

Figure 8: 1 million token needle-in-a-haystack. Comparison between Xiao et al. (2023) and our method on a needle-in-a-haystack task. The red cells show that attention sinks are incapable of retrieving tokens outside of the local window or early sink tokens. Our method is able to do so with $k = 10$ and extends to over 1 million tokens

## 5 RELATED WORK

### 5.1 EFFICIENT ATTENTION MECHANISM

The memory usage of the classic "self-attention" operation (Vaswani et al., 2017) grows quadratically with the input sequence length. This high demand for GPU memory impedes scaling LLMs to longer sequences, hindering many practical applications. To overcome this hurdle, recent works have developed various efficient alternatives to the attention operation. Here, we divide existing efficient attention mechanisms into two categories: one is "exact" attention (Dao et al., 2022), and the other is "approximate" attention (Beltagy et al., 2020; Han et al., 2023).

Exact attention algorithms strive to maintain near mathematical equivalence to the original attention, and the memory efficient approaches usually leverage efficient CUDA implementations (Dao et al., 2022), blockwise computation (Liu et al., 2023; Kwon et al., 2023), and sequence parallelization (Li et al., 2023). On the other hand, approximate attention algorithms save memory by reducing the size of the actual attention matrix at the loss of some of the original fidelity. A straightforward solution is sliding window attention (Beltagy et al., 2020), which assumes that local interactions dominate and long range dependencies can often be ignored. StreamingLLM (Xiao et al., 2024) discovered the "attention-sink" phenomenon and proposed a modified sliding window attention that alleviates the performance degradation in windowed attention. Other works explore the sparsity of attention matrices and propose different sparse attention operations to improve efficiency (Child et al., 2019; Qiu et al., 2020; Nawrot et al., 2024; Jiang et al., 2024). Our work falls into the approximate attention category; while similar ideas have been explored in previous work (Gupta et al., 2021; Klett & Ahle, 2024), we are the first to implement and run our top-k attention at a million token context-length scale.

### 5.2 SYSTEM SOLUTIONS TO LONG-CONTEXT IN LLMS

There is a long line of proposed system solutions to address the long-context problem in LLMs at inference. Flash Attention, and later extended to Flash-Attention 2, provides theoretical linear complexity over the sequence length, providing two to four times the memory savings over standard attention at inference time (Dao et al., 2022; Dao, 2024). vLLM's implementation of Paged Attention (Kwon et al., 2023) optimizes for throughput over many requests, but, like Flash Attention, this Paged Attention implements a block matrix multiplication algorithm that allows for memory savings at inference time. With a more retriever-like system, Klett & Ahle (2024) propose using sliding window attention to build a cache and k nearest neighbor (kNNs) using its utilization up to 16k context length. For KV cache management, Wu et al. (2022) alter the attention mechanism to memorize the internal representations of past inputs to acquire new skills and knowledge. Li et al. (2024) propose SnapKV, a method that reduces KV cache size without the need for fine-tuning by carefully selecting clustered important KV positions for each attention head reducing memory while increasing speed. Adnan et al. (2024) introduce Keyformer, which reduces the size of the cache by utilizing the sparsity in language models and only considering specific keys to decrease latency

and increase throughput. Singhania et al. (2024) construct a low-cost proxy to full attention using PCA, which informs the token subset for the attention computation. However, these solutions do not support context lengths past a few hundred thousand tokens. Thus, Liu et al. (2023) propose Ring Attention, which employs blockwise computation for the self-attention and feedforward operations, distributing long sequences over multiple devices and overlapping key-value block communication with the computation of attention. However, this method is computationally demanding, both in terms of memory usage and processing speed, as it requires substantial resources to manage the parallelization of large sequences across devices while maintaining communication overhead. To the best of our knowledge our approach is the first to achieve inference on million token context windows on a single commodity GPU.

## 6 CONCLUSION

We have demonstrated the capability of a top-$k$ attention mechanism to operate at the million token scale on a single GPU. In addition, our investigation of attention distributions across layers points to future directions where the choice of $k$ can be adapted across layers. We achieve sublinear complexity and evaluate at over 95% accuracy on common benchmarks while using only 1% of the context length on average in the attention block. This exploitation of attention sparsity opens up new directions for efficient and viable solutions to long-context in language models.

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

## A APPENDIX

### A.1 FULL RULER RESULTS

### A.2 DISTRIBUTION OF ATTENTION SCORES

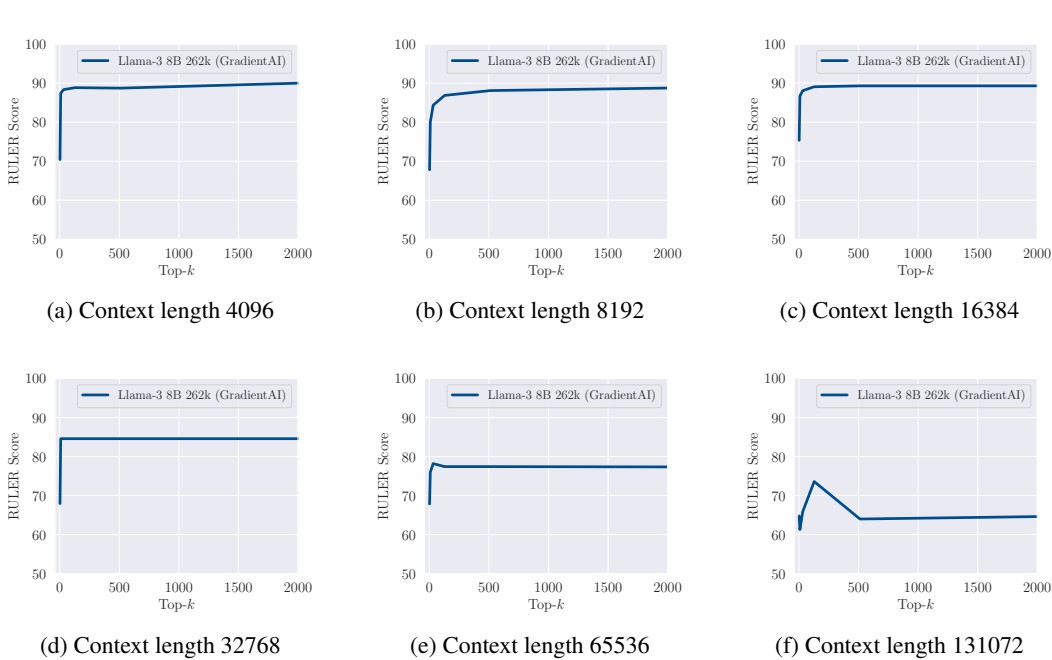

(a) Context length 4096    (b) Context length 8192    (c) Context length 16384

(d) Context length 32768    (e) Context length 65536    (f) Context length 131072

Figure 9: Results for RULER over various context lengths.

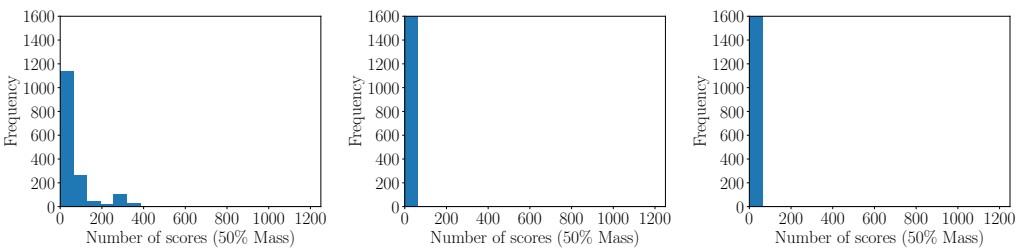

Figure 10: **Only a few number of scores or tokens are required to make up** $50\%$ **of the probability mass of a row in the attention matrix.** We analyze the number of attention scores that correspond to $50\%$ of the probability mass for generating the next token. Each point is the number of scores of the last 'row' from the attention matrix that make up $50\%$ of the probability mass. There are 1600 points comprised from the 50 datapoints and 32 heads. On the **left**, we plot the histogram for the first layer in the network, the **center** corresponds 16th layer, and **right** corresponds to the last layer. The sequence length of each sequence of 4000 tokens. However, we plot the range from 0 to 1250 on the x-axis as no ways of the three plots exceeds this range.

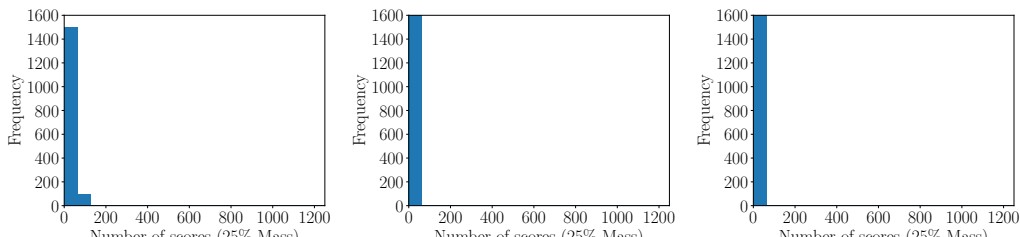

Figure 11: **Only a few number of scores or tokens are required to make up** $25\%$ **of the probability mass of a row in the attention matrix.** We analyze the number of attention scores that correspond to $25\%$ of the probability mass for generating the next token. Each point is the number of scores of the last 'row' from the attention matrix that make up $25\%$ of the probability mass. There are 1600 points comprised from the 25 datapoints and 32 heads. On the **left**, we plot the histogram for the first layer in the network, the **center** corresponds 16th layer, and **right** corresponds to the last layer. The sequence length of each sequence of $4000$ tokens. However, we plot the range from 0 to 1250 on the x-axis as no ways of the three plots exceeds this range.

## A.3   LAYER-WISE $k$: 8096 CONTEXT LENGTH

| 1st layer | other layers | score | 1st layer | other layers | score | 1st layer | other layers | score |
|-----------|--------------|-------|-----------|--------------|-------|-----------|--------------|-------|
| 2 | 2 | **70.5** | 33 | 1 | **49.3** | 32 | 1 | **49.4** |
| 8 | 8 | **83.1** | 132 | 4 | **74.1** | 32 | 2 | **70.8** |
| 32 | 32 | **85.3** | 528 | 16 | **83.7** | 32 | 4 | **77.8** |
| 128 | 128 | **87.2** | 2112 | 64 | **86.6** | 32 | 6 | **82.1** |
| 512 | 512 | **88.5** | full | 264 | **87.9** | 32 | 8 | **82.7** |
| 2048 | 2048 | **89.4** | full | 1850 | **89.8** | 32 | 12 | **83.4** |
| Full | Full | **89.3** | full | full | **89.3** | 32 | 16 | **84.4** |
| | | | | | | 32 | 24 | **84.6** |
| | | | | | | 32 | 32 | **85.3** |

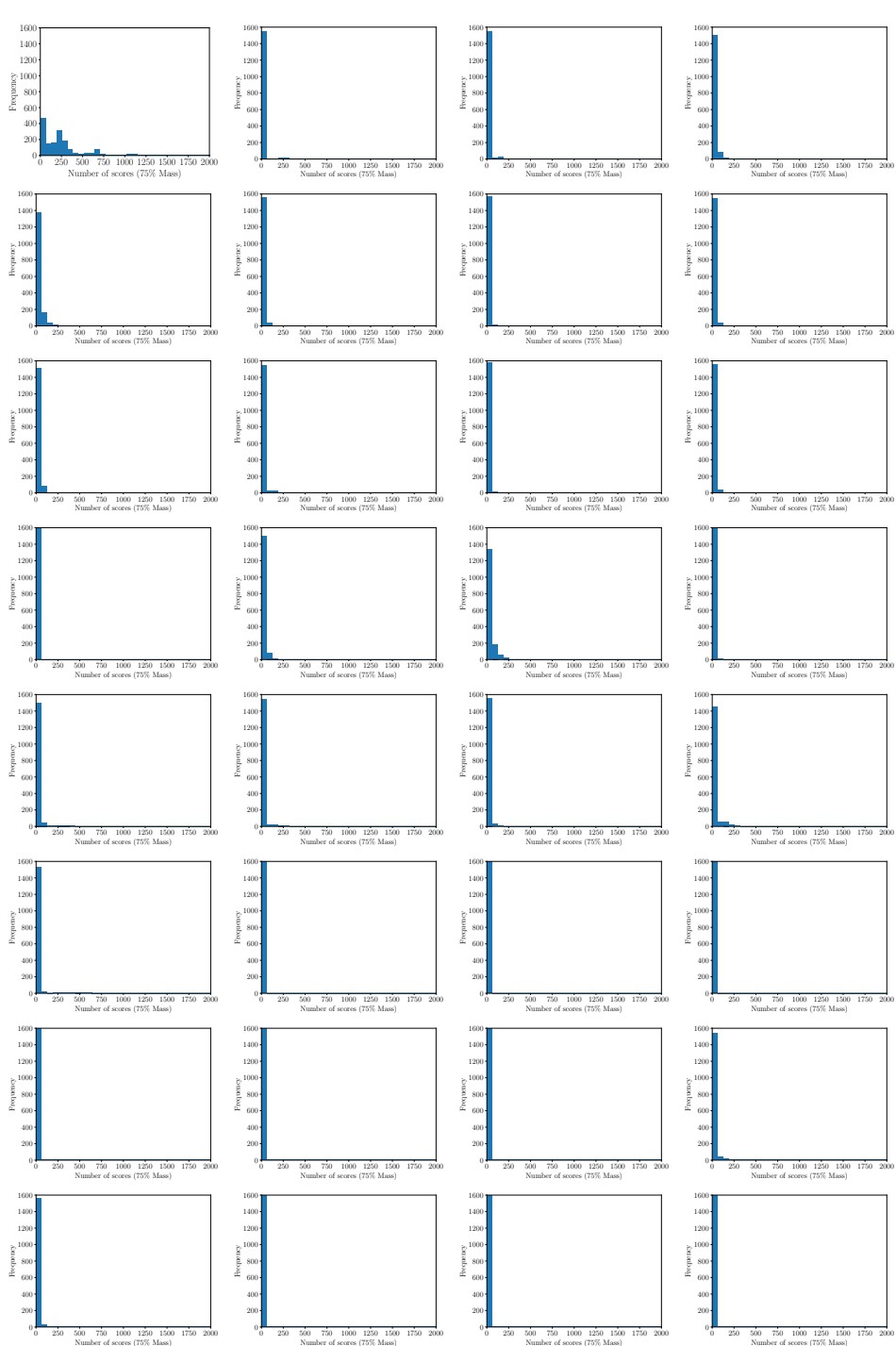

Figure 12: All 32 layers are plotted in order, where the top row represents layers 1, 2, 3, and 4 and last row represents layers 29, 30, 31, and 32.

