# OpenReview forum: "Running Huge Context Windows On Tiny GPUs"
_ICLR.cc/2025/Conference — Submitted to ICLR 2025_

### Official Review · Reviewer_kPdN · 2024-10-31

**Soundness:** 3
**Presentation:** 3
**Contribution:** 2
**Rating:** 6
**Confidence:** 4

**Summary:**

The paper proposes a way of reducing the computational requirements of attention blocks during inference in an existing pretrained LLM without requiring modifications to the neural architecture or additional training, and does so with a minimal degradation of it's capabilities. While the proposed algorithm is an approximation of the full attention, it's soundness is backed with observations and empirical evidence.
Furthermore, the implications of using efficient kv-caching during inference is also fully explored, which allows this method to be used as a drop-in addition to the attention algorithm used in most transformer LLMs.

**Strengths:**

- The problem is well motivated and very important. Runtime drop-in efficiency improvements to LLMs is preferable, especially without requiring additional training because pretraining is expensive and a single architecture might not fit all use cases. Some might prefer the accuracy of full attention while others require the speed of approximate attention. In both cases, the same pretrained model can be used, massively reducing pre-training compute requirements.
- Describes the implementation of a top-k kv-cache. This method can be easily adapted for use within a kv-cache accelerated inference framework.
- Solid and sound evals using AlpacaEval, Open LLM Leaderboard and RULER datasets. Testing both short and long context capabilities of top-k attention.

**Weaknesses:**

- Top-k attention as it stands by itself is not a novel algorithm, many papers have proposed it in the past, but not in the context of transformer LLM inference with kv-caching.
- A lack of comparison with other approximate attention algorithms with regards to benchmark scores and speed. (Why should we use this method rather than other more proven methods?)
- No table or numbers showing the compute and memory improvements. A paper focusing on inference efficiency should at least provide a few numbers in a table or graph showcasing the potential compute/memory savings.

**Questions:**

- How does this method scale to bigger models? If I understand correctly, the memory and compute savings should be larger and larger given that bigger transformer LLMs have more attention layers in general?
- In the paper, the author(s) notes a similarity to RAG, but why is there no comparison in the benchmarks? RAG is currently much more proven and highly efficient. It is currently unclear in the paper whether a naive top-k attention is really superior to it in practice without actual benchmarks.

---

> ### Author Response · Authors · 2024-11-21
> **Reply to Reviewer kPdN**
>
> Thank you for your insightful comments and questions. We answer them below.
>
> ### 1. Why should we use this method rather than other more proven methods?
> In long context modeling, quality can drop as context length grows, while simultaneously attention approximations must be made to fit contexts into GPU memory. Our solution, Top-k, removes the quality tradeoff to replace it with a vector database.
>
> Attention based on true top-k score selection is of vastly higher quality than approximate attention mechanisms. Attention Sink have known failure cases for Needle In A Haystack, as can be shown in figure 8 on page 9 of our newest draft. Attention Sink is incapable of retrieving information outside of its local window or it's early sink tokens, whereas Top-k is able to do so out to over 1 million tokens with k=10.
>
> Additionally, there are stark differences between true Top-k and approximate Top-k as reported in a recent paper, Loki[1]:
>
>    |                 | Hellaswag | TQA   | Winogrande | ARC   | MMLU  | Average |
>    |-----------------|-----------|-------|------------|-------|-------|---------|
>    | Full Attention  | 79.17     | 43.89 | 72.93      | 53.24 | 62.19 | 62.28   |
>    | Top-k Attention | 78.57     | 44.18 | 72.85      | 51.96 | 61.39 | 61.79   |
>    | H2O             | 70.79     | 30.84 | 50.12      | 32.85 | 30.39 | 43.00   |
>    | Loki            | 69.42     | 42.13 | 50.36      | 34.64 | 44.50 | 48.08   |
>
> Top-k attention maintains 95% quality with a 1% context budget, while H2O and Loki require over 30% context for the same threshold. Our performant ratio also holds at extremely long contexts.
>
> ### 2. Can you provide a table or graph showcasing the potential compute/memory savings?
> The primary challenge with long context inference is that the KV cache quickly becomes too large to fit into GPU memory. Top-k attention solves this by offloading the KV cache to a vector database, leaving a negligible memory footprint on the GPU. This is what enables 1-million token context lengths on a 16 GB GPU with an 8 billion parameter model.
>
> The peak GPU memory consumption Flash Attention and Top-k attention can help illustrate this:
>
> |                 | Peak GPU memory consumption |
> |-----------------|-----------------------------|
> | Flash Attention: | $2 * L * H_{kv} * D * N$   |
> | Top-k:          | $H_q * D * k$               |
>
> $N$: Context length
> $D$: Head dimension
> $H_q$: Number of Query heads
> $H_{kv}$: Number of Key-Value heads
> $L$: Number of layers
>
> With Top-k attention, only a small subset of the cache is moved to the GPU for each attention computation and the memory requirements scale with $k$ rather than with the context length ($N$).
>
> For a concrete comparison of memory requirements between Top-k and full attention, where $k$ is chosen as 1% of the total context length:
>
> | Context Length | Top-k (k=1%)   | Flash Attention     | Ratio (Memory Increase)
> |----------------|----------------|-----------|-------------|
> | 4,000          | 0.33 MB        | 0.52 GB   | 1575x
> | 32,000         | 2.62 MB        | 4.19 GB   | 1599x
> | 100,000        | 8.19 MB        | 13.11 GB  | 1600x
> | 1,000,000      | 81.92 MB       | 131.07 GB |  1600x
>
> Top-k attention presents a theoretical compute advantage because the vector database supports sublinear search time for a given query.
>
> ### 3. How does this method scale to bigger models?
> Top-k attention's GPU memory advantage scales differently depending on how the model is made larger. If the model is made deeper, Top-k attention becomes even more advantageous because it's GPU memory requirements do not include layers as a scaling factor, as shown in the table above. If the model is made wider, the scale of efficiency advantage will remain the same since both Top-k and competing attention methods include the model dimension equally in their theoretical memory requirements.
>
> ### 4. Why is there no comparison to RAG?
> Given that Top-k attention can operate at high fidelity over ultra-long document stores, we have occasionally wondered how long context models in general compare to RAG methods. While there are some trivial ways to show failure cases in RAG models (use a query that exceeds the context length of the retrieval model), a proper comparison between RAG and long context models deserves its own paper.
>
> Thank you for your thoughtful suggestions and the effort you put in to improving the quality of this work. We put significant effort into making improvements and running additional experiments based on your comments. Also, we were pleased that noted the ease with which this could be dropped into existing frameworks. Given our responses and updates, we would appreciate it if you would consider raising your score. We would love to continue the discussion and get this paper improved to the point that the results can be shared with and analyzed by the broader community.
> [1]: [Loki: Low-rank Keys for Efficient Sparse Attention](https://arxiv.org/abs/2406.02542)

---

> ### Comment · Reviewer_kPdN · 2024-11-22
>
> Thank you for running the additional experiments, I have adjusted the score accordingly. I think it would be also worthwhile to include these experiments and additional discussion about implications regarding compute/memory complexity in the paper.

---

### Official Review · Reviewer_UR9P · 2024-11-01

**Soundness:** 1
**Presentation:** 1
**Contribution:** 1
**Rating:** 3
**Confidence:** 5

**Summary:**

The paper introduces a KV cache compression method designed to enable longer context generation for large language models (LLMs) on memory-constrained GPUs. It maintains a comprehensive KV cache database on CPU memory and selectively transfers a top-k subset to GPU memory for efficient attention calculation. The method is evaluated through attention score analysis, a hyper-parameter
𝑘
k ablation study, and tests on huge context generation.

The results show over 95% accuracy on standard benchmarks while utilizing only 1% of the total context length, demonstrating significant memory efficiency.

**Strengths:**

Relevance: The paper addresses the critical and timely challenge of enabling long context generation in large language models under resource constraints, an essential topic for efficient LLM inference.

Effectiveness and Simplicity: The proposed method offers a simple yet effective approach to offloading KV caches, making it practical for implementation while delivering significant memory efficiency.

**Weaknesses:**

1. Lack of Comparative Analysis: The paper does not provide a comparison with related or concurrent works on KV cache sparsity, such as H2O and "Model Tells You What to Discard," missing an opportunity to contextualize its contributions within the broader research landscape. Additionally, there is an absence of detailed system-level performance metrics, such as latency, throughput, and supported context length, which are crucial for understanding the practical efficiency of the method.

2. Latency Concerns: The use of PCIe for data transfer between CPU and GPU may introduce significant latency overhead, which is not addressed in the paper.

3. Limited Dataset Scope: The evaluation does not include reasoning datasets (e.g., GSM8K, MATH), potentially overlooking the method’s impact on tasks that require strong reasoning capabilities.

4. Limited Acceleration Capability: The method can only accelerate the decoding stage but not the prefill stage. This limitation means that running the model on a single GPU with long context lengths is infeasible unless the prefill stage is computed using more powerful machines or other algorithms.

**Questions:**

1. Comparison with Other KV Cache Sparsity Works: The reviewer questions how this method compares with existing KV cache sparsity approaches. While the paper claims that extracting less than 1% of tokens still yields a high accuracy, other methods like Attention Sink, H2O, and "Model Tells You What to Discard" experience significant accuracy drops when reducing the KV cache to less than 30%. The reviewer seeks clarity on what makes this approach more effective.

2. Lack of Latency/Throughput Justification: The reviewer points out that the paper does not provide latency or throughput measurements. They emphasize the importance of system-level performance analysis, as highlighted in the weaknesses section.

3. Performance on Reasoning and Math Datasets: The reviewer asks about the method’s performance on challenging datasets, such as reasoning tasks (e.g., GSM8K) and math datasets (e.g., MATH), to evaluate how well the approach generalizes to tasks requiring strong reasoning capabilities.

---

> ### Author Response · Authors · 2024-11-21
> **Reply to Reviewer UR9P**
>
> ### 1. How does this method compares with existing KV cache sparsity approaches?
> It turns out that attention based on true top-k score selection is vastly superior to cache sparsity methods like H2O, which can be thought of as approximate top-k methods. In the recent Loki paper[1] they compare their method and H20 to true top-k and the results are stark:
>
>    |                 | Hellaswag | TQA   | Winogrande | ARC   | MMLU  | Average |
>    |-----------------|-----------|-------|------------|-------|-------|---------|
>    | Full Attention  | 79.17     | 43.89 | 72.93      | 53.24 | 62.19 | 62.28   |
>    | Top-k Attention | 78.57     | 44.18 | 72.85      | 51.96 | 61.39 | 61.79   |
>    | H2O             | 70.79     | 30.84 | 50.12      | 32.85 | 30.39 | 43.00   |
>    | Loki            | 69.42     | 42.13 | 50.36      | 34.64 | 44.50 | 48.08   |
>
> The same stark differences can be seen in simple Needle In A Haystack evaluations for methods like Attention Sink. We have added a figure to our newest draft -- figure 8 on page 9 -- demonstrating this. The red cells show that Attention Sink (StreamingLLM) is incapable of retrieving information outside of its local window or it's early sink tokens, whereas Top-k Attention (IVF Index) is able to do so out to 1 million tokens of context and beyond:
>
> Two of the main contributions of this paper are demonstrating that the high fidelity of Top-k attention shown in the table above hold for contexts at the million-token scale, and also quantifying exactly how sparse you can go while still maintaining some desired threshold of quality. As you mentioned, we found that Top-k attention can maintain 95% of full attention quality with a 1% context budget, while methods like H2O and Loki require more than 30% to reach this same threshold. The fact that this ratio holds true at ultra-long contexts is what makes true Top-k attention so promising.
>
> ### 2. Can you provide latency or throughput measurements?
> Here are our current inference time results. These were benchmarked on a 32 core AMD EPYC-7313 CPU with a 16GB NVIDIA RTXA4000 GPU using a Llama-3-8B model. Times are reported in seconds/token:
>
> |     N | Flash Attention | Attention Sinks | Top-k (k=10) |
> |------:|:----------------|:----------------|-------------:|
> |  1024 | 0.274           | 0.540           | 0.628        |
> |  2048 | 0.275           | 0.545           | 0.723        |
> |  4096 | 0.271           | OOM             | 0.872        |
> |  8192 | OOM             | OOM             | 1.259        |
> | 16384 | OOM             | OOM             | 1.932        |
> | 32768 | OOM             | OOM             | 3.872        |
>
>
>
> ### 3. What is the method's performance on reasoning tasks such as GSM8K and MATH?
> The favorable quality to sparsity ratio that we see on other benchmarks also holds for reasoning-focused benchmarks such as GSM8K. For a 2% context budget ($k$=56), Top-k attention achieves 90% of full attention quality. Here are results for increasing $k$, evaluating GSM8K with 5-shot in-context examples:
>
> | $k$ | GSM8K Score |
> |-----|-------------|
> | 4   | 6.52 |
> | 6   | 17.97 |
> | 10  | 28.89 |
> | 14  | 35.63 |
> | 24  | 41.02 |
> | 32  | 42.15 |
> | 48  | 44.81 |
> | 56  | 45.49 |
> | Full| 50.64 |
>
> We also see a similar trend in the MATH dataset. Here we downloaded the Llama 3.1 8B model from huggingface, and ran the LM Eval harness on the minerva_math task with varying k. This task is a four-shot variant of the MATH Benchmark. Our results are here:
>
> | $k$ | Minverva Score |
> |-----|----------------|
> | 20  | 0.1622 |
> | Full| 0.17 |
>
> We recover the performance of the full attention for our model with much smaller k. The context lengths of these examples vary in size but are around 1024. We note that our absolute performance matches that reported by the [OpenLLM Leaderboard](https://huggingface.co/datasets/open-llm-leaderboard/meta-llama__Meta-Llama-3.1-8B-details)
>
>
> Thank you for taking time to make this a better paper. It is encouraging that you find this method to be effective and practical for implementation. Based on your comments we put significant effort into revising and running additional experiments. We would appreciate it if you would consider raising your score in light of our response. We would love to continue the discussion and get this paper improved to the point that the results can be shared with and analyzed by the broader community.
> [1]: [Loki: Low-rank Keys for Efficient Sparse Attention](https://arxiv.org/abs/2406.02542)

---

> > ### Author Response · Authors · 2024-11-24
> >
> > Here are some updated results on the minerva math benchmark. As you can see, our method achieves similar results as full-k attention using a fraction of the total attention scores.
> >
> > |   Groups   |Value |   |Stderr| topk |
> > |------------|-----:|---|-----:|------|
> > |minerva_math|     0|±  |     0|  1   |
> > |minerva_math|     0|±  |     0|  2   |
> > |minerva_math|0.0432|±  |0.0029|  4   |
> > |minerva_math|0.1058|±  |0.0043|  8   |
> > |minerva_math|0.1272|±  |0.0046|  10  |
> > |minerva_math|0.1362|±  |0.0047|  12  |
> > |minerva_math|0.1492|±  |0.0049|  16  |
> > |minerva_math|0.1622|±  | 0.005|  20  |
> > |minerva_math|0.1628|±  | 0.005|  28  |
> > |minerva_math|0.1680|±  |0.0051|  32  |
> > |minerva_math|0.1628|±  |0.0051|  36  |
> > |minerva_math|0.1616|±  | 0.005|  40  |
> > |minerva_math|0.1634|±  |0.0051|  44  |
> > |minerva_math|0.1640|±  |0.0051|  48  |
> > |minerva_math| 0.170|±  |0.0052| full |
> >
> > Thanks again for your helpful comments! In light of our responses, would you be willing to increase the score on your review?

---

> > > ### Comment · Reviewer_UR9P · 2024-12-02
> > >
> > > Thank you for the response. I have the following questions and concerns:
> > >
> > > Top-k Selection vs. H2O:
> > > Can you explain why the top-k selection achieves such significant improvements in model accuracy compared to H2O? Your method retains only about 10 tokens in the KV cache, which is approximately 0.5-1% of the original KV cache, yet it demonstrates near-lossless performance. This seems counterintuitive, and further clarification would be helpful.
> > >
> > > Settings of KIVI and Loki:
> > > Could you provide the specific settings used for KIVI and Loki in your experiments? A detailed explanation would improve the transparency and reproducibility of your results.
> > >
> > > Handling Prefill Attention Scores for Very Long Contexts:
> > > How do you address the attention scores generated during the prefill phase for very long contexts? In Section 2 of the comment section, you claim that your method supports a sequence length of 32,768 for an 8B model. However, this seems unfeasible with a plain implementation, even with KV cache compression. This raises concerns about potential implementation issues in your approach.
> > >
> > > Overall, while your results are impressive, the above points require further clarification to validate the claims made in the paper.

---

> > > > ### Author Response · Authors · 2024-12-03
> > > > **Response to Comment by Reviewer UR9P**
> > > >
> > > > Thank you very much for your comments and help in improving the paper. We answer your questions below.
> > > >
> > > > **Top-k Selection vs. H2O**
> > > >
> > > > *Can you explain why the top-k selection achieves such significant improvements in model accuracy compared to H2O? Your method retains only about 10 tokens in the KV cache, which is approximately 0.5-1% of the original KV cache, yet it demonstrates near-lossless performance. This seems counterintuitive, and further clarification would be helpful.*
> > > >
> > > >
> > > > The difference between top-k attention and H2O is that our method chooses a different KV cache subset for each token. We store the entire KV cache of the context in a vector database so that every token can efficiently choose the subset it finds most useful for the attention computation. In contrast, H2O evicts tokens from the KV cache at every step of generation, meaning that future generation steps lose access to tokens that were not important previously. If a token in the kv cache hasn't contributed much of the total attention score during previous token generations, then it will be evicted in an H2O cache. Our method excels because what is important for the generation of one token could be very different for the next token. Even so, H2O is still an impressive method for achieving high inference throughput.
> > > >
> > > > **Settings of KIVI and Loki**
> > > >
> > > > *Could you provide the specific settings used for KIVI and Loki in your experiments? A detailed explanation would improve the transparency and reproducibility of your results.*
> > > >
> > > > The table we showed to compare top-k against H20 was from a recent paper, Loki [1]. With the Loki method, they first reduce the dimension of the keys in the kv cache and then perform top-k attention (without a vector database). Loki has two hyperparameters: d, the dimension of the reduced kv cache, and k, the number of vectors they select during each attention computation. In this table, d was selected to be 25% of the original embedding dimension, and k was selected to be 25% of the context length. For the experiments in Loki, they chose the size of the H20  cache to be 25% of the context length, evenly split between the local sliding window and the “heavy hitters.” They also provide results for exact top-k without the dimensionality reduction, where the top-k percentage they select is also 25% of the context length.
> > > >
> > > > KIVI quantizes the vectors in the kv cache to reduce memory. We see this as a complementary method that could be used in tandem with top-k attention, but have not run comparisons against it.
> > > >
> > > > **Handling Prefill Attention Scores for Very Long Contexts**
> > > >
> > > > *How do you address the attention scores generated during the prefill phase for very long contexts? In Section 2 of the comment section, you claim that your method supports a sequence length of 32,768 for an 8B model. However, this seems unfeasible with a plain implementation, even with KV cache compression. This raises concerns about potential implementation issues in your approach.*
> > > >
> > > > This is an important point, and we thank you for the opportunity to clarify our method. The use case we're imagining is an ultra-long context that remains fixed while the user makes repeated queries over it. There are a variety of ways you could prefill a long-context cache if it only needs to be done once. One option is to use multiple GPUs for this expensive, one-time operation. Another is to take more time to do the long context prefill on a high-memory CPU. You could also run a prefill using an approximate attention method. In our experiments, we prefill by utilizing Flash Attention [2], which provides some GPU memory savings, together with a routine that serializes the attention computation over heads to reduce peak memory cost.
> > > >
> > > > For the specific run we measure in with a sequence length of 32,768, we prefill the KV cache using the method mentioned above. Then, at decoding time, we first perform a top-k selection in the vector database, then move the selected keys to the GPU. In this instance, the attention layer uses 2.62 MB of GPU memory.
> > > >
> > > > **Conclusion**
> > > >
> > > > Thanks again for your help in producing this paper. We also believe the results are strong and would love to share this with the broader research community. If you feel that all of your concerns have been addressed we would kindly appreciate you changing your reviewer score.
> > > >
> > > > [1] [Loki: Low-rank Keys for Efficient Sparse Attention](https://arxiv.org/abs/2406.02542)
> > > >
> > > > [2] [FlashAttention: Fast and Memory-Efficient Exact Attention with IO-Awareness](https://arxiv.org/abs/2205.14135)

---

### Official Review · Reviewer_m5Jh · 2024-11-03

**Soundness:** 2
**Presentation:** 3
**Contribution:** 3
**Rating:** 5
**Confidence:** 4

**Summary:**

The paper presents a top-k attention mechanism aimed at reducing GPU memory usage to enable processing at the million-token scale on a single GPU. The authors highlight the sparsity of the attention matrix, where only a small subset of vectors meaningfully contribute to the attention scores, making much of the computation redundant. To address this, they propose a method that retrieves only the keys with the highest attention scores, leveraging a vector database in CPU memory. This allows attention computation to focus on the top contributors, while the retrieval is optimized using approximate k-nearest neighbor search, achieving sublinear complexity. This approach enables efficient long-context inference by offloading memory requirements to CPU without the overhead of full attention computation. Experiments show that the method achieves over 95% accuracy on common benchmarks, even with a relatively small top-k value.

**Strengths:**

The paper is well-written, with thoughtfully designed figures and diagrams that effectively illustrate the authors' top-k-based attention mechanism. The benchmarks clearly demonstrate performance variations with different top-k selections, and the design is well-motivated by the sparsity present in each layer of the attention scores—Figures 1, 2, and 3 showcase the feasibility of the proposed method convincingly.

While this approach shares similarities with RAG, it specifically focuses on top-k retrieval in the KV cache, preserving the model's zero-shot and reasoning capabilities. The method shows strong adaptability across transformer-based models and offers promising potential for extending supported context lengths, which could have a substantial impact across the field.

**Weaknesses:**

The idea of exploiting sparsity in the attention mechanism is not entirely novel, as this work primarily focuses on reducing the number of tokens attended to. However, a significant challenge remains in ensuring that the method achieves reasonable inference speed while reducing GPU memory consumption.

While the authors propose leveraging plentiful and affordable CPU memory with a vector database for k-nearest neighbor retrieval, there is no benchmarking or analysis to demonstrate the impact of this design on inference speed, raising concerns about the practical adaptability of the proposed approach.

Another limitation is the lack of empirical benchmarks on GPU memory consumption after applying the method. Additional results in this area would greatly enhance the paper.

**Minor issue:**  In Algorithm 2, the formatting for the Softmax and attention score  `S`  sections is inconsistent, which could lead to confusion. Highlighting the modifications would help readers identify the primary changes. Additionally, clarifying the relationships between  `K_cache`  and  `K_cache_gen`, as well as  `V_cache`  and  `V_cache_gen`, could improve comprehension.

**Questions:**

1. What is the GPU memory consumption after applying this method? Can we quantify the reduction based on the provided parameters?
2. How does the CPU memory vector database handle efficient searching? Are there any benchmark results available for inference speed?
3. What is the expected maximum context length for a given model with this method? Can we estimate it based on GPU memory constraints and model parameters?

---

> ### Author Response · Authors · 2024-11-21
> **Reply to Reviewer m5Jh**
>
> Thank you for taking the time to read and review our paper. We found your questions and comments insightful and answer them below.
>
> ### 1. What is the GPU memory consumption after applying this method?
> Here is the theoretical peak memory consumption of Top-k attention. For reference we compare with Flash Attention because it is the default attention implementation and has $O(N)$ memory complexity.
>
> |                 | Peak GPU memory consumption |
> |-----------------|-----------------------------|
> | Flash Attention: | $2 * L * H_{kv} * D * N$   |
> | Top-k:          | $H_q * D * k$               |
>
> $N$: Context length
> $D$: Head dimension
> $H_q$: Number of Query heads
> $H_{kv}$: Number of Key-Value heads
> $L$: Number of layers
>
> The peak consumption in popular implementations like Flash Attention or VLLM comes from maintaining the KV cache on GPU, whereas with Top-k attention the KV cache is offloaded to a vector database. Only a small subset of the cache is moved to the GPU for each attention computation, so the memory requirements scale with $k$ rather than with the context length ($N$).
>
> Here is a concrete comparison of memory requirements between Top-k and full attention, where $k$ is chosen as 1% of the total context length:
>
> | Context Length | Top-k (k=1%)   | Flash Attention     | Ratio (Memory Increase)
> |----------------|----------------|-----------|----------|
> | 4,000          | 0.33 MB        | 0.52 GB   |    1575x
> | 32,000         | 2.62 MB        | 4.19 GB   | 1599x
> | 100,000        | 8.19 MB        | 13.11 GB  | 1600x
> | 1,000,000      | 81.92 MB       | 131.07 GB |  1600x
>
> ### 2. How does the CPU memory vector database handle efficient searching?
> There is a large body of work on efficient searching over vector databases and specialized compute instances optimized for this type of work. Our implementation uses the FAISS library which we think is a sensible choice but was not optimized for this workload. We benchmarked our inference times on a 32 core AMD EPYC-7313 CPU with a 16GB NVIDIA RTXA4000 GPU using a Llama-3-8B model. Times are reported below in seconds/token and Flash Attention times are reported for reference:
>
> |     N | Flash Attention | Top-k (k=10) |
> |------:|:----------------|-------------:|
> |  1024 | 0.274           | 0.628        |
> |  2048 | 0.275           | 0.723        |
> |  4096 | 0.271           | 0.872        |
> |  8192 | OOM             | 1.259        |
> | 16384 | OOM             | 1.932        |
> | 32768 | OOM             | 3.872        |
>
> These runtimes could certainly be further optimized, but our method enables generation at long contexts on limited hardware where other methods degrade.
>
> ### 3. What is the expected context length for a given model with this method?
> Here is the formula for maximum context length given a specific GPU and a specific model configuation. This is an inversion of the formula listed in question 1 above. Here we choose $k_{budget}$ to be 1% of the total context length:
> |                 | Maximum context length      | Example for Llama-3-8B and RTXA5000 |
> |-----------------|-----------------------------|-------------------------------------|
> | Full Attention: | $(Mem_{GPU} - Mem_{model}) / (4 * L * H_{kv} * D)$ | $(24GB - 16GB)/(4 * 32 * 8 * 128)$ = 61,035 tokens |
> | Top-k:          | $(Mem_{GPU} - Mem_{model}) / (2 H_q * D * k_{budget})$ | $(24GB - 16GB)/(2 * 32 * 128 * 0.01)$ = 98m tokens |
>
> As shown above, there is no practical limit to the context length based on GPU memory for Top-k attention. In this regime the maximum expected context length becomes limited by the storage capacity of the vector database.
>
>
> Again, we'd like to thank you for the time you put in to helping sharpen and improve this publication. Your comments pointing out the formatting issues in Algorithm 2 were particularly helpful. We made a significant effort to revise our paper and run additional experiments in light of your feedback. We are pleased that you think this method of extending context length could have substantial impact, and would appreciate it if you would consider raising your score in light of our response. We would be happy to continue the discussion and get this paper improved to the point that the results can be shared with and analyzed by the broader community.

---

> > ### Comment · Reviewer_m5Jh · 2024-11-24
> >
> > Thank you for the detailed response; I appreciate the effort to explain and clarify my questions. While the authors addressed some of my concerns regarding memory consumption and context length, I remain unconvinced that the overhead of CPU-GPU communication can be mitigated to an acceptable level for LLM inference using the proposed method. It would be helpful if the authors could further demonstrate the inference speed of the proposed method, particularly at larger context lengths.

---

> > > ### Author Response · Authors · 2024-11-24
> > > **Reply Regarding Data Movement Time**
> > >
> > > I appreciate your comment and feedback on the rebuttal. While we were brainstorming, CPU-GPU transfer was a concern of ours as well. However, the truth is that the majority of the time of the computation is in fact spent doing the nearest neighbor search, and hardly any is spent in CPU-GPU transfer.
> > >
> > > To show this, I generated 5 tokens for 5 different contexts at varying context lengths using needle-in-a-haystack data with k=10. This k is warranted as it was able to achieve perfect results on the needle in a haystack test. For each run, I profiled the generation call using cProfile, and broke the timing down into three parts: parts spent in vector search, parts spent in movement (.to(device), .cpu(), and .cuda() calls), and parts spent elsewhere. I averaged the times over all tokens to get a per-token cost. The results are as follows:
> > >
> > > |      N |   Search Time (s) |   Move Time (s) |   Total Time (s) |   Other Time (s) |   % Time in Search |   % Time in Move |   % Time in Other |
> > > |-------:|------------------:|----------------:|-----------------:|-----------------:|-------------------:|-----------------:|------------------:|
> > > |   1024 |           0.18608 |         0.08556 |          0.57976 |          0.30812 |           0.32096  |         0.147578 |          0.531461 |
> > > |   2048 |           0.23748 |         0.06936 |          0.61704 |          0.3102  |           0.38487  |         0.112408 |          0.502723 |
> > > |   4096 |           0.23992 |         0.10236 |          0.62512 |          0.28284 |           0.383798 |         0.163745 |          0.452457 |
> > > |   8192 |           0.45844 |         0.11908 |          1.00044 |          0.42292 |           0.458238 |         0.119028 |          0.422734 |
> > > |  16384 |           0.59252 |         0.1728  |          1.23348 |          0.46816 |           0.480364 |         0.140091 |          0.379544 |
> > > |  32768 |           1.22588 |         0.27892 |          2.07076 |          0.56596 |           0.591995 |         0.134695 |          0.27331  |
> > > |  65536 |           2.69052 |         0.5204  |          4.05404 |          0.84312 |           0.663664 |         0.128366 |          0.20797  |
> > > | 131072 |           6.6396  |         0.95708 |          9.01344 |          1.41676 |           0.736633 |         0.106184 |          0.157183 |
> > > | 262144 |          15.006   |         3.99252 |         26.8767  |          7.87812 |           0.558329 |         0.14855  |          0.293121 |
> > >
> > > CPU-GPU transfer only contributes slightly to the overall time of the computation, and as the context length increases, more and more time is spent in the vector search. Indeed, since k is fixed, after the vector selection is performed, only k vectors need to be moved from the cpu to the gpu to perform the computation.
> > >
> > > In terms of accelerating our method, I would like to note again that we used off-the-shelf databases with minimal parameter tuning. We view this project as a proof-of-concept that enables generation at longer contexts with minimal GPU memory usage. There are optimizations that could surely be made to increase the speed of the vector search -- the operation which constitutes the vast majority of computation time -- including multithreading, multiprocessing and state-of-the-art database algorithms. In fact, a recent paper from database experts at microsoft research points towards ideas from vector databases that could be used to accelerate our method [1].
> > >
> > > I hope this response clarifies any issues you have, and I thank you once again for your insights on the paper. Given our response, do you see this leading to a change in our score?
> > >
> > > [1] [RetrievalAttention](https://arxiv.org/abs/2409.10516)

---

> ### Comment · Reviewer_m5Jh · 2024-11-24
>
> Thank you for providing additional benchmarking results! As noted, the majority of the time is spent on nearest neighbor search. I agree with the authors that the overall design could benefit from improved algorithms or novel search methods on the CPU side to reduce latency. However, I would also like to highlight that the proposed method may not yet be ready for production due to the potential for high latency in scenarios with long context lengths. I will raise my score to a 5 with more confidence.

---

### Author Response · Authors · 2024-12-04
**Summary of Rebuttal to Area Chairs**

Dear Area Chairs,

Thanks again for your service in making this conference happen. As the rebuttal period draws to a close, we'd like to summarize some of the questions raised by reviewers and highlight our responses and overall contributions.

**Comments by Reviewer m5Jh**

This reviewer requested clarification on the GPU memory cost for our method, which we provided. Our method takes significantly less GPU memory than current state of the art methods. Additionally, they were concerned about the cost of moving data between the CPU and GPU. We ran additional experiments to emphasize that the major cost of our method is in the vector database search, since the CPU-GPU movement is dependent on k which we take to be small. This search cost could be brought down significantly using advanced vector database and multithreading techniques, e.g those found in [1]. Additionally, we’d like to note that we view this method as a research work rather than a production-ready system.

**Comments by Reviewer UR9P**

This reviewer requested that we compare our results to other existing cache sparsity methods, like H20. The comparisons we provided demonstrated that top-k attention produces much higher fidelity generations than H20. The reviewer remains curious as to how our method could yield such significantly improved results, but we note that our method performing too well would be a strange reason to reject the paper. They also requested we run our method on math and reasoning benchmarks. We did this and observed the same results as the benchmarks in the paper, namely, that a fraction of the total context is needed to achieve baseline performance. We also shared with them latency benchmarks for our method.

**Comments by Reviewer kPdN**

This reviewer noted the strength of the evaluations in our paper, but requested a table specifying the theoretical memory savings of our method. We then provided the theoretical memory savings. Additionally, this reviewer wanted us to benchmark our method against RAG. In response, we argued that while RAG and long-context generation have potentially overlapping use cases, it is nontrivial to construct a fair comparison. We believe a proper comparison between long-context methods and RAG is interesting but out of scope for this paper.

**Concluding Remarks**

The rebuttal period contained fruitful discussions which enriched our paper and our arguments, to which we thank our reviewers for their time. We have added full needle-in-a-haystack evaluation plots to our paper which include our method achieving perfect accuracy on a 1m length context with only 16GB of GPU memory.

We want to make a strong appeal for getting the results of this paper in front of the broader research community. This is the first work that demonstrates and quantifies the generation quality that is possible with a top-k selection mechanism at attention time.

One of the critiques that we have gotten in this review process is that it is hard to understand why our method performs so much better on long contexts than previously proposed methods. We believe this reinforces the need to get more attention and research on this approach.

We hope you’ll consider accepting this paper in light of these discussions and improvements. Thanks again for your service.


[1] [ParlayANN: Scalable and Deterministic Parallel Graph-Based Approximate Nearest Neighbor Search Algorithms](https://arxiv.org/abs/2305.04359)

---

### Meta-Review · Area_Chair_4Dv3 · 2024-12-20

**Metareview:**

The paper proposes a method to enable LLMs to process very long contexts without exceeding GPU memory, by leveraging a sparse attention mechanism using a nearest neighbor search offloaded to CPU instead of GPU.
Unfortunately, several concerns remained about the paper. Main downsides were the high latency of nearest neighbor search, as well as incomplete comparisons to other approaches like H2O.'

We hope the detailed feedback helps to strengthen the paper for a future occasion.

**Additional Comments On Reviewer Discussion:**

The author feedback phase was useful as acknowledged by the reviewers. Some of the concerns however remained if the work is ready for the high bar of ICLR.

---

### Decision · Program_Chairs · 2025-01-22

Reject